# Bridging the Gap: Exploring Bronchopulmonary Dysplasia through the Lens of Biomedical Informatics

**DOI:** 10.3390/jcm13041077

**Published:** 2024-02-14

**Authors:** Jennifer Kim, Mariela Villarreal, Shreyas Arya, Antonio Hernandez, Alvaro Moreira

**Affiliations:** 1Division of Neonatology, Department of Pediatrics, University of Texas Health San Antonio, San Antonio, TX 78229, USA; kimj13@uthscsa.edu (J.K.); villarreal11@livemail.uthscsa.edu (M.V.); hernandezaj@uthscsa.edu (A.H.); 2Division of Neonatal-Perinatal Medicine, Dayton Children’s Hospital, Dayton, OH 45404, USA

**Keywords:** biomedical informatics, artificial intelligence, machine learning, bioinformatics, bronchopulmonary dysplasia, chronic lung disease

## Abstract

Bronchopulmonary dysplasia (BPD), a chronic lung disease predominantly affecting premature infants, poses substantial clinical challenges. This review delves into the promise of biomedical informatics (BMI) in reshaping BPD research and care. We commence by highlighting the escalating prevalence and healthcare impact of BPD, emphasizing the necessity for innovative strategies to comprehend its intricate nature. To this end, we introduce BMI as a potent toolset adept at managing and analyzing extensive, diverse biomedical data. The challenges intrinsic to BPD research are addressed, underscoring the inadequacies of conventional approaches and the compelling need for data-driven solutions. We subsequently explore how BMI can revolutionize BPD research, encompassing genomics and personalized medicine to reveal potential biomarkers and individualized treatment strategies. Predictive analytics emerges as a pivotal facet of BMI, enabling early diagnosis and risk assessment for timely interventions. Moreover, we examine how mobile health technologies facilitate real-time monitoring and enhance patient engagement, ultimately refining BPD management. Ethical and legal considerations surrounding BMI implementation in BPD research are discussed, accentuating issues of privacy, data security, and informed consent. In summation, this review highlights BMI’s transformative potential in advancing BPD research, addressing challenges, and opening avenues for personalized medicine and predictive analytics.

## 1. Introduction

Bronchopulmonary dysplasia (BPD) is a chronic lung disease primarily affecting extremely low gestational age neonates, with an annual incidence of 10,000–15,000 infants [1,2]. Recent years have witnessed changes in the definition and classification of BPD, where severity is now assessed based on the required respiratory support at 36 weeks postmenstrual age (PMA) [3]. The pathogenesis of BPD is intricate, stemming from a complex interplay of factors including developmental immaturity, injury, inflammation, and subsequent repair processes [4]. Notably, despite minimal improvement in BPD mortality rates since 2001, there has been a concerning rise in its overall incidence. This increase may be attributed to the growing number of neonates born at lower gestational ages [5]. As advancements in treatment strategies have led to enhanced survival rates among smaller and more premature infants, there is an urgent need for deeper exploration of the long-term morbidities faced by these vulnerable infants.

In recent years, significant strides in genetic engineering and molecular biology have paved the way for an upsurge in research investigating the pathophysiology of BPD [6]. While numerous transcription factors, genetic markers, and pathways have contributed valuable insights into BPD’s development, the challenge remains to pinpoint the most pivotal regulatory factors. Fortunately, with the remarkable progress achieved in the field of bioinformatics, researchers now have the capability to systematically assess tens of thousands of genes and proteins within the context of BPD [6]. This transformative capability streamlines the exploration of the intricate molecular mechanisms underlying BPD, offering a more comprehensive understanding of this complex condition.

## 2. Overview of Biomedical Informatics

Biomedical informatics has played a pivotal role in shaping medical innovation for decades [7]. Its roots can be traced back to the establishment of the American Association of Medical Record Librarians in 1928, which focused on the systematic organization and collection of medical records. This pioneering organization would later evolve into the influential American Health Information Management Association, a leading authority in managing health information in our nation’s healthcare landscape [7]. A significant resurgence of biomedical informatics occurred with the enactment of the 2009 American Recovery and Reinvestment Act, providing a monumental USD 19 billion investment in Health Information Technology. This transformative initiative championed the adoption of electronic health records and data collection practices, ushering in a new era of healthcare informatics [7].

Biomedical informatics, also referred to as biomedical and health informatics or just health informatics, is a broad discipline guided by technology that seeks to improve our understanding of and management of individual health, healthcare applications, public health, and medical research using health information [8]. The scope of biomedical informatics extends beyond medicine, encompassing fields such as computer science and biomedical engineering. Fundamentally, biomedical informatics revolves around three core hierarchical concepts: data, information, and knowledge. Data form the foundation of the hierarchy, but in their raw state, they are simply building blocks. For example, the data point “98.6” could be a patient temperature, an outdoor temperature, a radio station, a lab value, a distance between two points, a dollar amount, etc. Data, as elaborated in a *Journal of Biomedical Informatics* article [9], must undergo processing to acquire meaning, resulting in valuable information. This relationship is illustrated in Figure 1. The application of data and information to address pertinent questions ultimately leads to the generation of knowledge. In essence, biomedical informatics leverages computer processes and algorithms to organize data into medically relevant information, facilitating advancements in healthcare knowledge and beyond.

### 2.1. Key Concepts and Principles

Biomedical informatics has been defined as “the interdisciplinary field that studies and pursues the effective uses of biomedical data, information, and knowledge for scientific inquiry, problem solving, decision making, motivated by efforts to improve human health” [8]. Biomedical informatics should be differentiated from bioinformatics, the latter being a subset of the first and dealing only with data and information on the cellular and molecular level. Bioinformatics can be further divided into many subfields, including genomics, proteomics, and transcriptomics.

Biomedical informatics also stands apart from closely related fields like computer science, mathematics, statistics, and biomedical sciences, as each has its own distinct focus and limitations [9]. Computer science primarily emphasizes the development of efficient algorithms to solve problems but lacks the capacity to uncover the underlying meaning behind these results. Mathematics and statistics serve as invaluable tools within informatics, aiding in the identification of abnormal data patterns and features. However, their scope is limited to the recognition of anomalies, offering little insight into the causes behind these occurrences. Biomedical sciences, often mistaken for bioinformatics, revolve around devising solutions to biomedical challenges through the creation of devices or software meant to simulate biomedical processes. This field doesn’t center on data, its interpretation, or the knowledge derived from it [9]. Nevertheless, given the intricate complexity of the human body, analyzing individual body systems without acknowledging their interconnectedness still poses extraordinary difficulty for the field of biomedical informatics [9].

### 2.2. Role of Biomedical Informatics in Healthcare and Research

Knowledge-driven outcome research, a critical facet of healthcare, delves into how healthcare interventions and practices impact patient outcomes, cost utility, and effectiveness [10]. Biomedical informatics assumes a pivotal role in facilitating these research endeavors, bridging the gap between clinical insights and data-driven decisions. Biomedical informatics harnesses an array of information technology (IT) systems, encompassing both general and clinical platforms, such as PubMed, Electronic Health Records (EHRs), and data mining tools [10]. Moreover, it extends to specialized research IT systems, including disease models, electronic data collection mechanisms, and screening tools [10]. The expanding landscape of regional Health Information Exchange (HIE) organizations and comprehensive data repositories empowers researchers to conduct longitudinal or episodic queries tailored to specific areas of interest [11]. This newfound capability facilitates the retrieval of patient cohorts, rigorous hypothesis testing, and the identification of critical data trends, thereby unlocking a plethora of potential research and learning applications [11].

## 3. Navigating Challenges in BPD Research

Before the era of surfactant therapy and widespread use of antenatal steroids, bronchopulmonary dysplasia (BPD) manifested primarily in modestly preterm infants, resulting from lung injury and characterized by parenchymal inflammation and fibrosis [12]. However, the landscape of BPD has transformed in the contemporary era of ‘gentle’ ventilation practices and careful management to limit oxygen toxicity. The pathological features of what we now refer to as “new” BPD have evolved significantly, marked by alveolar simplification, decreased septation, and dysregulated pulmonary vascular development. These changes can be attributed to the arrest in lung development during the canalicular phase (16–26 weeks) and saccular phase (24–38 weeks) [13,14]. Consequently, the clinical presentation of BPD spans a spectrum of respiratory distress, including tachypnea and increased work of breathing, wheezing linked to the degree of airway narrowing and reactivity, and rales that correlate with the extent of pulmonary edema [15].

Understanding the shift in the pathological underpinnings of BPD is crucial in the modern neonatal care landscape. While ‘old’ BPD was a consequence of lung injury and inflammation, contemporary BPD is characterized by disruptions in the intricate phases of lung development. This evolution has implications for diagnostic approaches and therapeutic strategies. In this context, biomedical informatics can play a pivotal role by enabling the integration of vast and complex clinical data, such as imaging, genetic markers, and patient demographics, to better understand the multifaceted nature of BPD. Harnessing predictive analytics and machine learning, biomedical informatics can aid clinicians in early risk stratification, prognosis assessment, and personalized treatment planning, ultimately improving the outcomes of infants at risk for or affected by BPD”.

Moreover, BPD extends beyond its perinatal origins, with emerging research revealing a broader spectrum of implications. Infants with BPD face lifetime risks of pulmonary and cardiovascular complications, such as emphysema, chronic wheezing, and heart failure [16]. Neurological associations have also come to the forefront, with documented links to conditions like retinopathy of prematurity, developmental delay, and cerebral palsy [16]. Despite persistent efforts, prevention strategies and therapeutic interventions have failed to curb the escalating incidence of BPD or substantially improve its clinical trajectory and long-term consequences [2]. Consequently, there remains an imperative need for sustained research endeavors. This ongoing pursuit is essential to unraveling the intricacies of this multifaceted disease and unveiling contributing factors that underlie its persistence.

In summary, BPD research has evolved in tandem with advances in neonatal care, necessitating a shift in our understanding and approach to this complex condition. Biomedical informatics, with its capacity to integrate diverse clinical data and employ cutting-edge analytical tools, offers a promising avenue to improve diagnosis, treatment, and ultimately, the long-term outcomes of infants affected by BPD. However, the challenges posed by this multifaceted disease demand continued research efforts, spanning from basic science to clinical investigations, to address its complexity and reduce its burden on vulnerable neonates and their families.

### 3.1. Unraveling Risk Factors and Causes

A comprehensive understanding of BPD necessitates the exploration of factors contributing to its development, which can be chronologically categorized into antenatal, natal, and postnatal realms [17] (see Figure 2). 

Antenatal factors: These encompass genetic susceptibility, intrauterine growth restriction (IUGR), chorioamnionitis, maternal smoking, and pregnancy-induced hypertensive disorders. While chorioamnionitis itself is not a standalone BPD risk factor, its consequential sequelae, notably sepsis, may elevate the likelihood of BPD development [18]. Pregnancy-induced hypertensive disorders, including gestational hypertension, preeclampsia, and eclampsia, are associated with BPD due to their connection with IUGR [19].Natal factors: These chiefly revolve around gestational age and birth weight [17].Postnatal factors: BPD’s postnatal milieu includes oxidative stress and hyperoxia, mechanical ventilation, sepsis, patent ductus arteriosus (PDA), and respiratory microbial dysbiosis [17]. Hyperoxic oxidative stress is a well-known risk factor, stemming from the generation of free radicals that culminate in endothelial damage and tissue disruption, ultimately affecting the pulmonary alveolar-capillary membrane [20]. Volume-induced lung injury (VILI) is another significant risk factor, resulting from barotrauma associated with prolonged mechanical ventilation, leading to surfactant dysfunction and regional hypoxia [21]. Hemodynamically significant PDA has also been linked to BPD development, driven by increased pulmonary blood flow, heightened ventilator demands, and exacerbated lung inflammation [22,23].

In addressing BPD prevention, various strategies are employed, encompassing antenatal steroids, non-invasive ventilation, nasal continuous positive airway pressure (CPAP), and targeted oxygen saturation level.

### 3.2. Current Diagnostic and Treatment Approaches

Currently, treatment strategies for BPD are implemented at various intervals throughout the entire hospital course of premature infants. Early hospital course interventions include surfactant administration, Vitamin A, methylxanthines, and gentle ventilation strategies such as non-invasive respiratory support [24]. Middle-course interventions include PDA management/closure, systemic corticosteroids, and optimization of nutrition and fluid balance [24]. Late-course interventions include diuretic therapies such as furosemide and thiazides and airway modulation with inhaled therapies such as bronchodilators and inhaled corticosteroids [24]. At every stage, each intervention has benefits and risks, and each intervention’s overall impact and contribution to BPD prevention can be debated.

### 3.3. Unveiling Research Frontiers in BPD

An array of research prospects illuminates the uncharted territories of our understanding of BPD [4]. Key avenues for exploration encompass the following:Categorization and diagnosis: A profound need persists for refined methods of categorizing and diagnosing BPD. This entails delving deeper into its subtypes and associated clinical markers.Risk factors: A thorough investigation of contributing risk factors is essential. This encompasses a multifaceted analysis, spanning antenatal, natal, and postnatal factors, to decipher the intricate interplay leading to BPD.Delivery room interventions: Unanswered questions abound in the domain of delivery room interventions. Innovative approaches such as noninvasive surfactant administration techniques, cord clamping vs. cord milking, and the complexities surrounding antenatal consent warrant rigorous examination.NICU management: The neonatal intensive care unit (NICU) presents a rich landscape for exploration. The role of novel therapies, the identification of BPD susceptibility in the postnatal period, and the establishment of optimal fluid and electrolyte targets demand meticulous investigation.Pathological lung samples: An in-depth analysis of pathological lung samples offers invaluable insights into the underlying mechanisms of BPD. This avenue calls for comprehensive studies to unravel the cellular and molecular intricacies within afflicted lungs.Outpatient follow-up: Given the chronic complications faced by BPD patients, the optimization of treatment strategies in the outpatient follow-up setting is imperative. Long-term care protocols should be meticulously crafted to enhance patient outcomes and quality of life.

## 4. Biomedical Informatics Approaches for BPD Research

There are a variety of different forms of biomedical informatics, some of which will be outlined in the sections below and are represented in Figure 3. 

### 4.1. Data Collection and Management from Electronic Health Records

Each electronic health record (EHR) contains patient data that can be separated into two different categories: structured data, such as prescriptions, vital signs, and lab values, and unstructured data, such as imaging studies and clinical notes. One of the strengths of EHRs lies in the clarity and accessibility of healthcare data. These digital repositories contain comprehensive patient histories, encompassing clinical notes, laboratory results, radiological images, and treatment records, among others. This wealth of data, when effectively used, allows biomedical informaticians to provide critical insights into diseases, treatment outcomes, and population health trends. Rapid growth in the use of temporal EHR data opens the door to many new research possibilities, including more comprehensive risk stratification models, prediction of chronic disease progression, and earlier detection of adverse drug events [25].

The primary benefit of data clarity within EHRs is the standardization and organization of health information [26]. Clinical data are consistently formatted, minimizing ambiguity and errors that can occur in handwritten records. With standardized data elements, biomedical informaticians can efficiently analyze and compare data across different patients, medical facilities, and even geographic regions. This consistency not only improves the quality of research but also facilitates seamless data sharing and collaboration among healthcare professionals and researchers.

Moreover, EHRs offer real-time access to patient data, ensuring that biomedical informaticians work with up-to-date information [27]. This dynamic access to clinical records allows for timely interventions, supports evidence-based decision-making, and enables the development of predictive models. For instance, in the context of BPD, researchers can track the progression of the disease in neonates, monitor the effectiveness of interventions, and identify risk factors—all through the clarity and timeliness of EHR data.

Another strength is the ability to capture longitudinal patient data [28]. EHRs store patient information over time, allowing researchers to follow disease trajectories, assess treatment responses, and identify patterns that may not be apparent in a single snapshot of data. This longitudinal view is particularly valuable in chronic conditions like BPD, where the evolution of the disease and its long-term effects on patients can be tracked, ultimately leading to more comprehensive research and improved patient care.

Overall, the clarity of data obtained from EHRs serves as a cornerstone of biomedical informatics. It empowers researchers with standardized, organized, and timely information, fostering a deeper understanding of diseases like BPD. By optimizing the strengths of EHR data clarity, biomedical informaticians can drive innovation, enhance patient outcomes, and advance the field of healthcare research.

### 4.2. Machine Learning and Artificial Intelligence: Transforming Medicine

The inception of “artificial intelligence” (AI) traces back to John McCarthy, who coined the term in 1956, defining it as “the science and engineering of making intelligent machines” [29]. Early AI systems primarily comprised expert systems, relying on a repository of “if-then” statements. However, it swiftly became evident that tackling more intricate challenges demanded machines capable of data interpretation, learning, and informed problem-solving [29].

In the contemporary landscape, AI in its diverse forms, including machine learning (ML), among others, finds application across various domains. Machine learning, a prominent AI subset, hinges on exposing the machine to a training dataset [30]. Within ML, distinct subcategories emerge, encompassing reinforcement learning, deep learning, supervised learning, and unsupervised learning. Irrespective of the AI subtype employed, the medical arena witnesses a burgeoning array of applications. In medicine, AI systems often draw insights from patient EHRs and extensive imaging databases. A burgeoning focus on bronchopulmonary dysplasia (BPD) research leverages AI, primarily through various machine learning models. These models navigate the intricate landscape of clinical and genetic risk factors to discern infants with BPD within research populations. Notable instances of artificial intelligence in BPD research are eloquently delineated in Table 1 [31,32,33,34,35].

### 4.3. Genomics and Personalized Medicine

Transcriptomics, a pioneering field, harnesses cutting-edge technologies like microarrays and RNA sequencing to delve into the intricate molecular mechanisms governing bronchopulmonary dysplasia (BPD). Complementing this, proteomics and metabolomics ventures characterize distinct protein expressions and metabolites, often gleaned from samples such as amniotic fluid, urine, and lung fluid [24]. Recent strides in these domains have significantly enhanced our comprehension of BPD, shedding light on the intricacies of human lung development [24].

Intriguingly, current research hints at a robust hereditary component intertwined with BPD development. Genome-wide association studies (GWAS), however, have yet to yield substantial breakthroughs in the context of BPD. Wang et al.’s comprehensive genomics study, encompassing 1726 infants, was unable to pinpoint single nucleotide polymorphisms (SNPs) linked to BPD development or validate previously identified SNPs associated with the condition [36].

Contrastingly, alternative approaches have illuminated the pathophysiology of BPD. Leveraging GWAS-derived data, one study unveiled two novel pathways, miR-219 and CD44, with potential roles in the genetic predisposition to BPD. Moreover, it unearthed rare exome mutations that might disrupt pulmonary structure and function in the BPD milieu [36]. More recently, investigations into the airway microbiome of BPD patients have unveiled distinctive microbiome signatures, hinting at the possible influence of bacterial colonization in the respiratory tract on BPD development [37]. These multifaceted approaches collectively unravel the intricate molecular tapestry underpinning BPD.

### 4.4. Predictive Modeling and Simulation

Predictive modeling, a well-established mathematical abstraction, leverages known system components to generate predictions for elusive properties within a system [38]. While this concept has found great application in diverse fields like data science and computational biology, its foray into the realm of bioinformatics is relatively recent. In the context of bioinformatics, predictive modeling harnesses the synergy of omics data and patient information, weaving a predictive tapestry. This powerful tool thrives on input data drawn from biomedical informatics methodologies, rooted in the tangible realm of measurable and observable data. Its output unveils the enigmatic properties of interest or previously undisclosed clinical entities. Relationships between these variables come to light through data-fitting methodologies or provisional estimations.

The versatile realm of predictive modeling unfolds numerous applications, including the early detection of diseases, offering non-invasive alternatives to traditional diagnostics, identifying rare genetic diseases via single nucleotide polymorphisms, and forecasting disease trajectories [39,40]. An illustrative contemporary application resides in the modeling endeavors crafted to anticipate the impacts of SARScov2 transmission and control measures during the COVID-19 outbreak. Statistical and ordinary differential equations-based models have excelled in projecting the anticipated incidence peaks, hospitalization rates, and their temporal occurrence. Simulations stemming from susceptible and infected ODE models have emerged as invaluable tools, guiding decision-makers in implementing effective control measures and minimizing casualties within specific geographic regions [41,42,43].

Endeavors to apply predictive modeling using bioinformatic approaches (or omic technology) to BPD research have already begun. One study by Moreira et al. incorporated a peripheral blood transcriptomic signature in some of their models used for the prediction of BPD [44]. With data from a cohort of 97 neonates, the research team developed several predictive models, some relying on birthweight and gestational age, and other ML models that employed a grouping of five genes. The models using birthweight and gestational age accurately predicted BPD, with an AUC of 87.2% and 87.8%, respectively [44]. The four machine learning models using the five-gene signature yielded AUCs between 85.8% and 96.1% [44]. What is also at the forefront of current BPD research is the creation of endotypes, groupings of genes differentially expressed amongst neonates with known BPD [45]. One study described four endotypes clustered using ML with data from a genome-wide profile of 62 children. The endotypes were also shown to represent differences in gestational age, birthweight, and T helper 17 cell differentiation [45]. These advancements in BPD research show the promise that predictive modeling holds in the exploration of the complicated nature of BPD. 

### 4.5. Mobile Health

Conceived in 2003, the visionary concept of mobile health (mHealth) is encapsulated as “mobile computing, medical sensors, and communication technologies for healthcare” [46]. The landscape underwent a seismic shift in 2007 with the advent of smartphones, prompting the World Health Organization (WHO) to redefine mHealth as “a medical and public health practice supported by mobile devices, such as mobile phones, patient monitoring devices, personal digital assistants, and other wireless devices” [47].

The contemporary realm of mHealth unfolds a myriad of applications, exemplified by wearable devices that meticulously track vital signs, portable microscopes and probes seamlessly integrated with cell phones, and text messaging programs and applications meticulously crafted to promote healthy behaviors. Amid the tumultuous COVID-19 pandemic, new mHealth modalities emerged, encompassing digital contact tracing apps and vaccination passport applications, underscoring the boundless potential of these technologies.

The transformative impact of mHealth reverberates globally, transcending barriers to healthcare, especially in less developed countries grappling with transportation challenges and healthcare worker shortages. In a landmark 2011 global survey on eHealth (pertaining to healthcare services fortified by digital processes), the WHO reaffirmed its unwavering commitment to advancing the industry. Collaborating with the International Telecommunication Union (ITU), the WHO embarked on the creation of a National eHealth Roadmap Development Toolkit, a pivotal step in forging comprehensive eHealth strategies [46]. This resounding dedication finds robust support on a national level, with reputable sponsors including the National Institutes of Health (NIH), Fogarty, the National Cancer Institute, and other influential entities [48].

### 4.6. Biomedical Imaging Informatics

In recent years, there has been a remarkable surge in the application of artificial intelligence (AI) and machine learning (ML) techniques in medical imaging, and this trend is primed to make significant contributions to our understanding and management of BPD. One noteworthy area where BMII can have a transformative impact is in the early detection and prediction of BPD in premature infants through chest X-rays. By employing deep learning algorithms, researchers have been able to develop automated systems that can analyze and interpret chest X-rays to identify subtle patterns and anomalies indicative of BPD. For instance, a study conducted by Xing et al. demonstrated the feasibility of using deep learning to detect radiographic signs of BPD in neonatal chest X-rays [49]. This pioneering work showcases the potential for BMII to assist clinicians by providing rapid and accurate assessments of BPD risk, enabling timely interventions and personalized care strategies for at-risk infants.

In a recent study by Chen et al., deep learning algorithms were employed to classify children with congenital heart diseases using electrocardiogram images [50]. The neural network model was built on a dataset comprising approximately 65,000 cases. Remarkably, the model exhibited an area under the curve of over 90% when tested on external datasets. What makes this achievement even more impressive is that the algorithm’s ability to recognize cardiac diseases surpassed the diagnostic capabilities of pediatric cardiologists themselves. This study underscores the potential of biomedical imaging informatics in revolutionizing the field of neonatology. Such a tool, when tailored to the intricate clinical and imaging data associated with BPD, holds the promise of not only enhancing diagnostic accuracy but also offering invaluable insights into disease progression and treatment strategies for this critical neonatal condition”.

In conclusion, the integration of AI-driven image analysis with clinical data, as well as genomic information, opens new avenues for research and clinical practice. As BMII continues to evolve and gain momentum, its applications in BPD and other neonatal conditions are positioned to make substantial contributions to the field, enhancing the quality of care for our tiniest and most vulnerable patients.

## 5. Clinical Tangibles of Biomedical Informatics

Biomedical informatics has a promising role in neonatal research. Recent studies integrating biomedical informatics into research are summarized below in Table 2 [51,52,53,54,55]. The application of biomedical informatics to BPD encompasses several goals:Early diagnosis and risk stratification: Biomedical informatics enables the development of predictive models that can identify infants at the highest risk of developing BPD. By leveraging clinical data, genetic markers, and imaging, these models can provide early warnings to healthcare providers, allowing for targeted monitoring and interventions.Severity prediction: One of the primary objectives is to predict the severity of BPD in affected infants accurately. Biomedical informatics can aid in assessing the extent of lung damage, the need for respiratory support, and the duration of oxygen therapy. These predictions are invaluable in tailoring treatment strategies and resource allocation in neonatal intensive care units.Personalized treatment plans: Biomedical informatics can assist in tailoring treatment plans for individual patients based on their unique clinical profiles and genetic markers. Personalized interventions can help optimize outcomes and minimize potential complications associated with standard treatments.Long-term outcome prediction: Beyond the neonatal period, biomedical informatics can also contribute to predicting long-term outcomes in infants with BPD. This includes assessing the risk of chronic respiratory conditions, neurodevelopmental disorders, and cardiovascular complications and providing families and clinicians with valuable information for ongoing care.Identification of therapeutic targets: Biomedical informatics can aid in identifying potential therapeutic targets for BPD. By analyzing molecular pathways and genetic factors associated with the condition, researchers can pinpoint novel interventions and drug candidates that have the potential to mitigate lung damage and improve outcomes.Mobile health integration: Biomedical informatics aims to seamlessly integrate mobile health technologies and data into the management of BPD. This includes the development of smartphone applications, wearable devices, and remote monitoring systems that allow real-time tracking of vital signs, respiratory patterns, and other relevant health metrics in infants with BPD. The aim would be to provide both healthcare providers and caregivers with accessible, real-time data, enabling early detection of deteriorating health and facilitating timely interventions.Enhanced clinical decision support: Biomedical informatics seeks to improve the accuracy and effectiveness of clinical decisions related to BPD through the integration of advanced decision support systems within EHRs. These systems utilize patient-specific data, including medical history, diagnostic results, and treatment plans, to provide evidence-based recommendations to healthcare providers. The purpose would be to reduce diagnostic errors and optimize treatment strategies.

## 6. Successful Applications of Bioinformatics in BPD Research

Despite its relatively recent emergence, the field of biomedical informatics has catalyzed remarkable strides in BPD research. A compelling illustration comes from Stanford University School of Medicine, where researchers conducted exome sequencing of neonatal blood samples from fifty pairs of twins, half affected by BPD and half unaffected [56]. Employing functional genome approaches, they scrutinized isolated nonsynonymous mutations, unveiling their pivotal involvement in pulmonary structure and function. These findings underscore the significant contribution of both rare genetic variants and already-identified common variants to BPD pathogenesis.

In 2014, another study harnessed microarray gene expression data alongside genome-wide DNA methylation data, comparing preterm infants with BPD to their term counterparts [57]. The investigation unearthed 23 genes displaying differential methylation patterns between the two groups, with gene expression and methylation exhibiting an inverse correlation with normal lung development. Among these genes, notable candidates included those associated with detoxifying enzymes (Gstm3) and transforming growth factor-beta signaling (Bmp7). These revelations hint at the regulatory role of DNA methylation in governing normal versus aberrant alveolar septation [57].

### 6.1. Benefits and Limitations of Biomedical Informatics

Embracing the multifaceted realm of genomics, transcriptional profiling through DNA microarrays has emerged as a potent tool for deciphering biological systems. However, the inherent diversity across microarray platforms and protocols, inherent to different laboratories, has posed a formidable challenge to data reproducibility. These disparities can significantly hamper the comparability of gene expression analyses [58]. Notably, the Microarray Gene Expression Data Society has crafted a vital safeguard in the form of a specification checklist known as the Minimum Information About a Microarray Experiment (MIAME). MIAME endeavors to standardize research protocols and diminish variations among datasets, fostering more reliable outcomes [59].

Despite the extensive data accessible in EHRs, they are not without limitations. The magnitude of big data analysis necessitates a deep reservoir of domain-specific knowledge, coding ability, and understanding of complex medical concepts, which becomes essential to gleaning the true significance and implications of these findings [60]. The potential for harnessing tools like predictive modeling and artificial intelligence to address these challenges looms on the horizon, though much work is necessary before mass adoption (e.g., standardization of terms, values, and labs collected; development and validation across multiple centers) [61].

The WHO’s global survey on mobile health (“mHealth”) implementation found that competing health system priorities were the main barrier to greater implementation of mHealth. Since verified evidence regarding the benefits of mHealth is still lacking, other potential areas of research may take precedence. This lack of verified evidence has also led to the death of public knowledge regarding the possible application of mHealth and a lack of user knowledge of how to best utilize these electronic tools [48].

### 6.2. Potential for Future Research and Development

To unlock the full potential of deep learning algorithms in healthcare, the critical requirement is large-scale EHR visual datasets (e.g., X-rays, magnetic resonance images). A recent methodological review in 2022 shed light on a concerning trend—the majority of recent studies predominantly relied on a single dataset, with only a handful venturing into the utilization of two or more datasets. This highlights an imperative need for greater emphasis on exploring the transferability and generalizability of these models [25]. In light of these insights, the authors of this review proposed a visionary solution—the creation of a freely accessible global database, fostering multicenter validation of existing models. Such an initiative would significantly elevate the robustness and reliability of deep learning algorithms, facilitating their deployment across diverse healthcare settings.

Additionally, the concept of transfer learning emerges as a potent strategy, enabling the transfer of crowdsourced knowledge from one institution to another [62]. This approach holds the promise of expediting the adoption of successful models, catalyzing progress in healthcare AI, and ultimately enhancing patient care on a global scale.

## 7. Discussion

### 7.1. Ethical and Legal Issues Related to Biomedical Informatics in BPD Research

Like all new technologies, biomedical informatics has presented some ethical considerations. In order to properly analyze these considerations, it is important to recall the three pillars of ethics: autonomy, beneficence, justice, and nonmaleficence. With the exponential rise in social media popularity, many applications and websites have promoted access to medical information on a new level. Although this promotes patient autonomy, these “consumer informatics” carry the risk of spreading potentially lethal misinformation and misinterpretation of data [63]. Technological breakthroughs such as the Electronic Medical Records and Computerized Provider Order Entry systems have irrefutably improved clinical efficiency, interdisciplinary collaboration, and patient education [64].

Nevertheless, one must consider the pillar of beneficence: If the ideal use of biomedical informatics systems provides significant benefits, do physicians have a moral obligation to utilize them and address their safety flaws? One related topic of debate pertains to whether secondary data analysis is inherently included within the realm of patient consent for primary analysis. Some justify that the use of patient data for research purposes helps to serve the public good, while others argue that patients typically do not explicitly agree to the documentation of their clinical encounters for any purpose other than their own healthcare [65].

Another important ethical consideration is the pillar of justice. The very nature of artificial intelligence and deep learning algorithms may unintentionally lead to the development of biases related to secondary factors such as race, ethnicity, gender, etc. These risks demand greater responsibility and scrutiny from physicians. Some have even suggested incorporating anti-bias features into the code itself, a promising possibility [63].

Finally, legal liabilities must be considered: does the vendor assume legal responsibility for faulty software or incorrect recommendations? Vendors often include the clauses “indemnity and hold harmless” in their contracts to free them from this responsibility. However, many would argue that they have a moral obligation to take responsibility for their errors, just as users have a responsibility to use good judgment when utilizing these tools.

### 7.2. Future Directions for BPD Research Using Biomedical Informatics

A 2016 American College of Medical Informatics (ACMI) winter symposium tackled an ongoing debate on the relationship between biomedical informatics and data science [66]. Rather than existing in separate planes of study, the group concluded that biomedical informatics and data science should co-exist synergistically. Consequently, they suggested that biomedical informaticians should also be trained in data science and vice versa. They concluded that the two fields should attempt to make contributions to each other, despite their differences. A coordinated plan should therefore be implemented to promote inclusiveness between these two interdisciplinary communities and their relevant stakeholders. Additional curricular frameworks must also be designed to assist in the mastery of the critical concepts common to both domains. Another issue involves delineating the data ownership and responsibilities of financial funding. Both the biomedical informatics and data science communities will face challenges related to high NIH administrative overhead costs, a challenge that may be overcome by establishing the National Library of Medicine (NLM) as a common homebase for these two communities [10].

Another obstacle in the field of biomedical informatics is its translation and application to real-world situations. The novelty of this field is partially responsible for this challenge, given poor public understanding of the applicability of computing solutions, inadequate infrastructure, and a generalized resistance to change among certain societies. Collaboration with information technology companies has improved the standards for storing and transferring patient information while maintaining privacy and generalizability for secondary analysis [62].

Transfer learning (the transfer of knowledge from one area to another), unsupervised learning (methods to create models without having complete sets of labeled data), and causal interference (methods to infer causal knowledge from data) present opportunities for progress in the field of artificial intelligence (AI) research [67]. Advancements in these areas would also likely help to address the challenges of data scarcity and generalizability [62].

## 8. Conclusions

The field of biomedical informatics is quickly gaining traction in the scientific research world. Unlike other fields of study, biomedical informatics incorporates information, data, and, most importantly, meaning to potentially shed light on any missing pieces of information. One example is the pathophysiology of BPD, which is a significant contributor to the morbidity and mortality of small, premature infants. Thanks to recent advances in biomedical informatics, physicians are now one step closer to fully understanding the intricacies behind BPD development. However, there exists much room for improvement. Increasing the number of and the standardization of large-scale EHR datasets available for analysis, unifying the fields of biomedical science and data science, and improving the translatability of biomedical principles are just a few improvements that could drastically advance the field. Given the novelty of biomedical informatics, it is imperative to improve the general public’s understanding and awareness of its applicability within the medical and scientific communities.

## Figures and Tables

**Figure 1 jcm-13-01077-f001:**
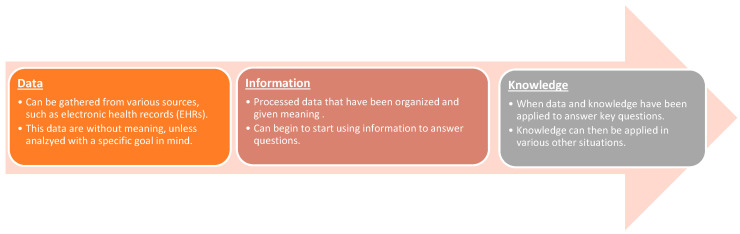
Tenets of biomedical informatics.

**Figure 2 jcm-13-01077-f002:**
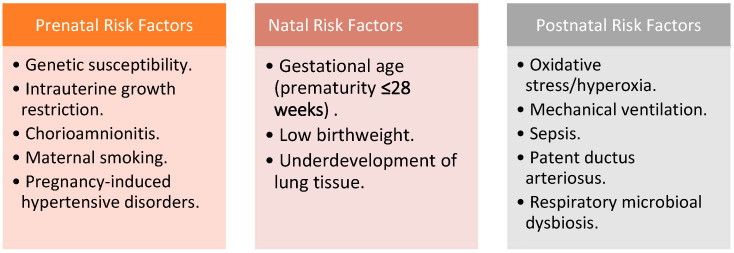
Risk Factors for BPD.

**Figure 3 jcm-13-01077-f003:**
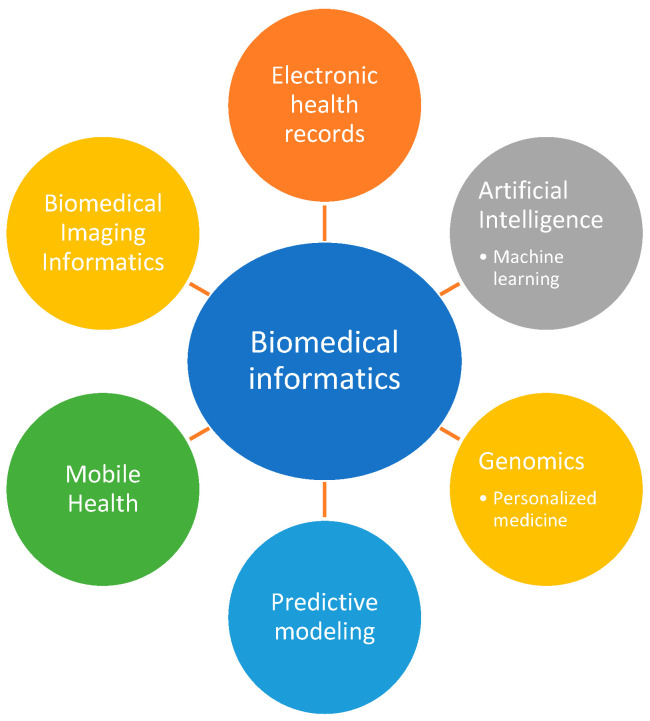
Breakdown of biomedical informatics.

**Table 1 jcm-13-01077-t001:** Applications of AI for BPD research.

Author(s)	Geographic Location	Year of Publication	No. of Neonates Involved in the Study	Inclusion Criteria	Artificial Intelligence Model Used	Overall Outcome
Verder et al. [31]	Denmark	2021	61 premature infants: 26 with BPD 35 without BPD	Infants born prematurely who were between 24 and 31 gestational weeks (previously enrolled in a past observational study by authors).	Support Vector Machine (SVM)	Authors predicted BPD in premature infants using surfactant treatment, spectral data from samples of gastric aspirate, birth weight, and gestational age. The specificity and sensitivity of the test were 91% and 88%, respectively. Compared to other diagnostic tests, the test developed by the authors provides results more quickly, allowing for treatment to begin as soon as possible.
Lei et al. [32]	Western China	2021	648 premature infants: 149 with BPD 499 without BPD	Infants with congenital anomalies, premature infants that died, and premature infants with incomplete data before diagnosis of BPD were excluded from the study.	Random Forest	The authors formulated a model using six variables to predict BPD (area under the receiver operating curve [AUC] of 0.929). The variables used included total oxygen inhalation time, first PCO_2_, first MAP (mean airway pressure), gestational age, gross birth weight, and first FIO_2_.
Dai et al. [33]	Shanghai, China	2021	245 premature infants: 131 with BPD 114 without BPD	Premature infants (<32 weeks gestational age) who required supplemental oxygen on their first day of post-natal life and were admitted to the NICU. Those with significant diseases, those that died, and those that refused to participate/undergo sequencing were excluded from the study.	Unsupervised Machine Learning and Least Absolute Shrinkage and Selection Operator	Using machine learning, the authors built predictive models that incorporated clinical and genetic data (from identified risk genes). They were successful in predicting both BPD and severe forms of BPD (AUC 0.915 and AUC 0.907, respectively). The accuracy of these models exceeded that of models only incorporating clinical data.
Morag et al. [34]	Israel	2021	208 premature infants: 40 with mild BPD, 16 with moderate-severe 152 without BPD	Infants born prematurely between January 2012 and August 2015 and completed a follow-up exam. Those with major congenital anomalies were excluded from the study.	Random forest, logistic regression classifier, gradient boosting classifier, XGBoost, and ExtraTree classifier	The authors used machine learning to assess the impact of several environmental factors on inhaler use in children who were born with BPD. Frequency of inhaler use served as a measure of long-term respiratory outcomes for these individuals. Inhaler use was more significant (*p*= 0.0014) in those with a greater number of risk factors and moderate-severe BPD. The identification of these factors (cigarette smoking, allergies, etc.) could allow for early intervention by the doctor and parents to limit exposure.
Ochab and Wajs [35]	Poland	2016	109 premature infants: 46 diagnosed with BPD after the fourth week of life	Infants born prematurely that weighed less than or equal to 1500 g and were admitted into the NICU by their second day of life.	Support Vector Machine	The authors developed an expert system that allowed for the sequential inclusion of clinical parameters (presence of patent ductus arteriosus, birth weight, etc.) to yield a model capable of producing the most accurate prediction of BPD. They compared both logistic regression and SVM models and found that models that incorporated more than 7 parameters had more accuracy using an SVM approach (accuracy of 83.29%)

**Table 2 jcm-13-01077-t002:** Sample studies integrating biomedical informatics in neonatology research.

Author	Publication Year	PMID	Objective	Theme	Outcome
Lavilla et al. [51]	2021	34550843	Measure organ dysfunction changes by gestational age and among extremely preterm infants	Descriptive	Neonatal sequential organ failure assessment scores, calculated hourly, effectively discriminated between survival and non-survival from the first day of life.
Hum et al. [52]	2014	25024755	Develop and implement a clinical decision support (CDS) tool aimed at enhancing antibiotic prescribing in the NICU	CDS	The CDS tool was activated for 22% of patients prescribed antibiotics. Summarized culture results and antibiotic recommendations were described as the most useful features of the tool. Widespread use was hindered by multiple systemic changes (e.g., new EHR, changes to antimicrobial testing).
Campbell et al. [53]	2022	35157950	Validate an AI software to detect retinopathy of prematurity (ROP)	Digital imaging	Strong correlation between the ROP vascular severity score and diagnosis of stage among ophthalmologists.
Das et al. [54]	2024	38195661	Profile the blood of very preterm babies during episodes of sepsis and identify immune signatures	Bioinformatics	Amphiregulin (AREG), gene involved in tissue repair, was identified as a gene that becomes dysregulated in neonates with bacterial sepsis.
Greenberg et al. [55]	2022	35728925	Create an accurate online estimator for predicting bronchopulmonary dysplasia BPD or death	Predictive	Authors developed multinomial regression models and translated them into a web-based tool for estimating BPD risk in extremely preterm infants.

## Data Availability

Not necessary as this is not original study with data, but instead a review article.

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
