# Peer review of "Bridging the Gap: Exploring Bronchopulmonary Dysplasia through the Lens of Biomedical Informatics"

_jcm, 2024, doi:10.3390/jcm13041077_

Round 1

Reviewer 1 Report

Comments and Suggestions for Authors

In the review paper: Bridging the Gap: Exploring Bronchopulmonary Dysplasia Through the Lens of Biomedical Informatics, the authors provided evaluation of application of innovative strategies in analyzing biomedical data of bronchopulmonary dysplasia in order to get better insight into the disease, diagnostics, prevention and treatment. The authors gave a detailed overview about the application of biomedical informatics in data analysis which can be eployed in personalized medicine. 

The review is comprehensive and interesting and can have value in diagnostic and management of BPD.

Major revision:

In order to emphasis the importance of advanced processing of medical data and analysis of BDP outcomes I recommend brief addition of clinical and pathological features of BPD, including histopathological changes associated with the disease, as well as the consequences of BPD in children and adolescences and the risk for other pulmonary diseases in children and/or adults. 

Author Response

Comments 1: “Major revision:

In order to emphasis the importance of advanced processing of medical data and analysis of BDP outcomes I recommend brief addition of clinical and pathological features of BPD, including histopathological changes associated with the disease, as well as the consequences of BPD in children and adolescences and the risk for other pulmonary diseases in children and/or adults. “

Response 1: Thank you very much for your suggestions. In response to your comment, we have included the text in quotations below. This text can be located on page 4 of the newly submitted manuscript. Again, thank you so much for you comments and for allowing us to improve on our manuscript. 

“Before the era of surfactant therapy and widespread use of antenatal steroids, bronchopulmonary dysplasia (BPD) manifested primarily in modestly preterm infants, resulting from lung injury and characterized by parenchymal inflammation and fibrosis [12]. However, the landscape of BPD has transformed in the contemporary era of 'gentle' ventilation practices and careful management to limit oxygen toxicity. The pathological features of what we now refer to as 'new' BPD have evolved significantly, marked by alveolar simplification, decreased septation, and dysregulated pulmonary vascular development. These changes can be attributed to the arrest in lung development during the canalicular phase (16-26 weeks) and saccular phase (24-38 weeks) [13,14]. Consequently, the clinical presentation of BPD spans a spectrum of respiratory distress, including tachypnea and increased work of breathing, wheezing linked to the degree of airway narrowing and reactivity, and rales that correlate with the extent of pulmonary edema [15].

Understanding the shift in the pathological underpinnings of BPD is crucial in the modern neonatal care landscape. While 'old' BPD was a consequence of lung injury and inflammation, contemporary BPD is characterized by disruptions in the intricate phases of lung development. This evolution has implications for diagnostic approaches and therapeutic strategies. In this context, biomedical informatics can play a pivotal role by enabling the integration of vast and complex clinical data, such as imaging, genetic markers, and patient demographics, to better understand the multifaceted nature of BPD. Harnessing predictive analytics and machine learning, biomedical informatics can aid clinicians in early risk stratification, prognosis assessment, and personalized treatment planning, ultimately improving the outcomes of infants at risk for or affected by BPD."

Moreover, BPD extends beyond its perinatal origins, with emerging research revealing a broader spectrum of implications. Infants with BPD face lifetime risks of pulmonary and cardiovascular complications, such as emphysema, chronic wheezing, and heart failure [16]. Neurological associations have also come to the forefront, with documented links to conditions like retinopathy of prematurity, developmental delay, and cerebral palsy [16]. Despite persistent efforts, prevention strategies and therapeutic interventions have failed to curb the escalating incidence of BPD or substantially improve its clinical trajectory and long-term consequences [2]. Consequently, there remains an imperative need for sustained research endeavors. This ongoing pursuit is essential to unravel the intricacies of this multifaceted disease and unveil contributing factors that underlie its persistence.

In summary, BPD research has evolved in tandem with advances in neonatal care, necessitating a shift in our understanding and approach to this complex condition. Biomedical informatics, with its capacity to integrate diverse clinical data and employ cutting-edge analytical tools, offers a promising avenue to improve diagnosis, treatment, and ultimately, the long-term outcomes of infants affected by BPD. However, the challenges posed by this multifaceted disease demand continued research efforts, spanning basic science to clinical investigations, to address its complexity and reduce its burden on vulnerable neonates and their families.”

Reviewer 2 Report

Comments and Suggestions for Authors

The manuscript entitled “Bridging the gap: exploring bronchopulmonary dysplasia through the lens of biomedical informatics” has been reviewed. The authors point out some new ideas and fields for the research and diagnosis of BPD. The manuscript has made quite a clear description and discussion about the application of biomedical informatics in current BPD research. There are some suggestions for the authors:

1.     It appears that the content of biomedical informatics is diverse. I would suggest the authors present a more detailed categorization about the biomedical informatics so the readers are able to know what they are looking for.

2.     The initial description of BPD by Dr. Northway mentioned about the appearance of the patients’ Chest X ray film. In this article, the image of Chet X ray film has not been mentioned. Is image study not included in the biomedical informatics? If the image study result is included, how to deal with the data?

3.     If we apply the biomedical informatics to BPD research, what are the ultimate goal, for example, prediction of severity or outcomes, early treatment, or only for better diagnosis? Could the authors give us some examples?

4.     What about the duration of data collection?

Besides the above questions, the manuscript looks promising.

Author Response

Comments 1: “It appears that the content of biomedical informatics is diverse. I would suggest the authors present a more detailed categorization about the biomedical informatics so the readers are able to know what they are looking for.”

Response 1: Thank you very much for your suggestions. In response to your comment, we have expanded upon the categorization of biomedical informatics. This is seen in the highlighted text on pages 2 and 3 of the manuscript.

Comments 2:The initial description of BPD by Dr. Northway mentioned about the appearance of the patients’ Chest X ray film. In this article, the image of Chet X ray film has not been mentioned. Is image study not included in the biomedical informatics? If the image study result is included, how to deal with the data?”

Response 2: Thank you for your feedback. This is addressed in the new section: 4.6 Biomedical imaging informatics on pages 11-12 (highlighted text).

Comments 3: “If we apply the biomedical informatics to BPD research, what are the ultimate goal, for example, prediction of severity or outcomes, early treatment, or only for better diagnosis? Could the authors give us some examples?”

Response 3: The reviewer’s comments regarding the applications of biomedical informatics to BPD are addressed in the new section: 5. Clinical Tangibles of Biomedical Informatics and further highlighted in the new Table 2, located on pages 12-13.

Comments 4: “What about the duration of data collection?”

Response 4: Clarification on the duration of data collection is included in the highlighted text in section 4.1 Data Collection and Management from electronic health records located on page 7.

The authors wish to once again thank you for your constructive feedback and for the opportunity to improve upon our manuscript.